# Interplay between MicroRNAs and Oxidative Stress in Neurodegenerative Diseases

**DOI:** 10.3390/ijms20236055

**Published:** 2019-11-30

**Authors:** Julia Konovalova, Dmytro Gerasymchuk, Ilmari Parkkinen, Piotr Chmielarz, Andrii Domanskyi

**Affiliations:** 1Institute of Biotechnology, HiLIFE, University of Helsinki, 00014 Helsinki, Finland; julia.konovalova@helsinki.fi (J.K.); dmytro.gerasymchuk@helsinki.fi (D.G.); ilmari.parkkinen@helsinki.fi (I.P.); 2Institute of Molecular Biology and Genetics, NASU, Kyiv 03143, Ukraine; 3Department of Brain Biochemistry, Maj Institute of Pharmacology, Polish Academy of Sciences, 31-343 Krakow, Poland

**Keywords:** microRNA, oxidative stress, ROS, translation regulation, neurodegeneration, Alzheimer’s disease, Parkinson’s disease, Huntington’s disease, ALS

## Abstract

MicroRNAs are post-transcriptional regulators of gene expression, crucial for neuronal differentiation, survival, and activity. Age-related dysregulation of microRNA biogenesis increases neuronal vulnerability to cellular stress and may contribute to the development and progression of neurodegenerative diseases. All major neurodegenerative disorders are also associated with oxidative stress, which is widely recognized as a potential target for protective therapies. Albeit often considered separately, microRNA networks and oxidative stress are inextricably entwined in neurodegenerative processes. Oxidative stress affects expression levels of multiple microRNAs and, conversely, microRNAs regulate many genes involved in an oxidative stress response. Both oxidative stress and microRNA regulatory networks also influence other processes linked to neurodegeneration, such as mitochondrial dysfunction, deregulation of proteostasis, and increased neuroinflammation, which ultimately lead to neuronal death. Modulating the levels of a relatively small number of microRNAs may therefore alleviate pathological oxidative damage and have neuroprotective activity. Here, we review the role of individual microRNAs in oxidative stress and related pathways in four neurodegenerative conditions: Alzheimer’s (AD), Parkinson’s (PD), Huntington’s (HD) disease, and amyotrophic lateral sclerosis (ALS). We also discuss the problems associated with the use of oversimplified cellular models and highlight perspectives of studying microRNA regulation and oxidative stress in human stem cell-derived neurons.

## 1. Introduction

Neurodegenerative diseases, such as Alzheimer’s (AD), Parkinson’s (PD), Huntington’s (HD) disease, and Amyotrophic Lateral Sclerosis (ALS), are devastating and currently incurable conditions causing severe cognitive and/or motor impairments predominantly in aged people [1,2]. The incidence of age-related neurodegeneration is expected to increase due to aging population and increased life expectancy in the developed countries. Alzheimer’s disease (AD) and other dementias are estimated to affect up to 50 million people worldwide [3]. Another 10 million patients are suffering from Parkinson’s disease (PD), which occurs in ≈2% of people over 70 years of age [4]. To develop curative therapies for neurodegenerative diseases, it is crucial to elucidate molecular mechanisms regulating neuron survival and degeneration.

Oxidative stress has been implicated in predisposing neurons to death either directly or indirectly as a consequence of mitochondrial dysfunction, pathological protein aggregation, specific neurotransmitter (dopamine) metabolism, inflammation, or deregulation of antioxidant pathways [5,6,7,8,9,10]. The brain is particularly susceptible to oxidative stress due to high oxygen consumption (reflecting high ATP demand) and the reliance on mitochondrial activity, intracellular calcium, and a relatively weak endogenous antioxidant defense, among other reasons [11,12]. Reactive oxygen species (ROS) cause oxidative damage to proteins, lipids, and nucleic acids, compromising critical cellular functions and activating cell death pathways [13]. Oxidative stress and oxidative damage are commonly observed in different neurodegenerative diseases and, therefore, therapies aiming to reduce cellular ROS levels may offer neuroprotective treatments for multiple neurodegenerative conditions. However, attempts to treat neurodegenerative diseases with antioxidant drugs have mostly been unsuccessful, in part, due to insufficient blood–brain barrier penetration, short treatment duration, or incorrect timing of therapy application [13,14,15]. Alternative therapeutic interventions may aim to counteract oxidative damage by stimulating endogenous neuronal antioxidant defense pathways [16]. In this review, we explore the concept of targeting specific microRNAs regulating or regulated by these pathways as a strategy to protect neurons in neurodegenerative diseases.

MicroRNAs are short regulatory RNA molecules which affect translation and stability of their mRNA targets by guiding RNA-induced silencing complex (RISC) predominantly to 3′ untranslated region (UTR) [17,18]. MicroRNAs are predicted to regulate the activity of about a half of all protein coding genes, reducing fluctuations in protein expression [19,20].

MicroRNAs are expressed as precursor hairpins which undergo sequential processing in the nucleus and cytoplasm by specific protein complexes containing ribonucleases Drosha and Dicer; mature functional microRNAs are then loaded to Argonaute family protein Ago2, a central component of the RISC complex [18]. MicroRNAs are critical for neuronal functions both during development and in the adult brain [21,22]. Loss of mature microRNA functions by genetic deletion of Dicer or Ago2 is embryonic lethal [23,24], whereas deletion of Dicer during embryogenesis severely impairs neuronal development [21,25,26,27,28,29]. Inducible deletion of Dicer in postnatal Purkinje cells and in adult forebrain and dopamine neurons causes their progressive loss and severe behavioral phenotypes [30,31,32,33,34]. While some neuronal populations survive Dicer deletion, their functions are clearly affected [35,36,37,38]. Similarly, loss of Ago2 in adult neurons is dispensable for their survival, but it nevertheless affects neuronal functions resulting in a behavioral phenotype [39]. MicroRNAs have been implicated in modulation of neuronal signaling by regulating neuronal excitability, dendritogenesis, local translation in dendritic spines, and neurotransmitter release [21,40,41,42]. Age and disease-related downregulation of the microRNA biogenesis pathway in adult neurons can lead to changes in their survival, functions, and connectivity. Inhibition of Dicer activity and resulting changes in microRNA expression levels have been observed in aging and in neurological and neurodegenerative diseases [30,42,43,44,45,46,47,48,49,50]. Deregulation of microRNA biogenesis is causing cellular stress and, vice versa, increased stress causes deregulation of microRNA biogenesis, creating a vicious cycle leading to eventual cell death [51,52,53]. In line with this hypothesis, stimulation of microRNA biogenesis is neuroprotective in mouse models of ALS and PD [30,52,54].

A small number of microRNAs can regulate hundreds of transcripts and may enable a crosstalk between different cellular pathways [55,56]. For example, several microRNAs can be targeting many genes involved in antioxidant defense pathways [53]. Modulating the levels of a relatively small number of microRNAs which regulate the oxidative stress response in neurons may therefore alleviate pathological oxidative damage and have neuroprotective activity. However, it is not trivial to identify such microRNAs and their target genes in aged and degenerating neuronal populations. Below, we review the current literature addressing the interplay between oxidative stress and microRNAs in major neurodegenerative diseases.

## 2. Alzheimer’s Disease

Dementia is estimated to affect more than 50 million people worldwide with the prognosis of doubling in the next 20 years [57]. AD, which is an irreversible neurodegenerative disorder affecting both cognition and emotional behavior of affected persons (usually at the age of 65 and older) [58], is considered to be the cause of 50–75% of all dementia cases with no effective treatment to stop or slow down the disease [59,60].

Despite intensive studies, the real causes of AD development are still not clear. Extracellular accumulation of amyloid-β (Aβ) peptides and hyperphosphorylation of the microtubule-associated Tau protein are the main hallmarks of AD development at the molecular and cellular level leading to the accumulation of senile plagues and neurofibrillary tangles, respectively. Mutations in *PSEN1*, *PSEN2*, *APP* genes, variants of *APOE* gene, and posttranscriptional modifications of AD-associated proteins can also contribute to the development of this neurodegenerative disease. Taken together, these changes result in synaptic loss, neuronal cell death, and cognitive impairment reviewed in [61,62].

According to numerous studies, microRNA contribute to the development of AD regulating accumulation of Aβ peptides and Tau phosphorylation [63,64,65,66,67,68]. However, accumulation of insoluble protein aggregates is not the only, and, possibly, not the main pathological process driving AD progression. Oxidative stress is of particular importance for AD development as it causes chronic inflammation at the early stages of neurodegeneration, which leads to mitochondrial dysfunction, oxidative damage of nucleic acids, changes in genes expression, and abnormal modifications of lipids and proteins [69]. Oxidative stress causes both up- and downregulation of different microRNAs and, conversely, many microRNAs can regulate oxidative stress response [70] (Figure 1).

Li et al. demonstrated that soluble Aβ peptides (sAβ) known to generate ROS [71] reliably induced expression of miR-134, miR-145 and miR-210. In the same study, expression of miR-107 was markedly reduced, supporting a bilateral effect of sAβ-induced ROS on microRNA expression [72]. Decreased levels of miR-107 is associated with early stages of AD progression. This microRNA directly targets BACE1 mRNA encoding β-secretase enzyme that processes APP to Aβ peptides [73]. In AD patients with the APOE4 genotype, decreased levels of miR-107 have been demonstrated along with the increased production of Aβ peptides. Accumulation of Aβ induced oxidative stress in APOE4 leads to the deregulation of the *TP53* gene. In addition to its role in cancer, p53 protein (encoded by a *TP53* gene) can be involved in cell death in AD patients with upregulation at the early stages of the disease and downregulation during neurodegeneration [74]. Previously, p53 mutations that may be associated with oxidative stress were observed in AD patients and AD animal models [75,76]. Since miR-107 is downregulated in cell lines with mutated p53 [77], p53 mutations and accumulation of Aβ may result in the decrease of miR-107 levels in AD patients. Moreover, 8-oxo-2′deoxyguanosine RNA modifications caused by oxidative stress can serve as an additional factor of decreasing miR-107 levels [78]. Levels of another microRNA, miR-186, are decreased through aging. This microRNA targets 3′UTR of BACE1 and is implicated in the mitigation of the oxidative stress effects in AD pathogenesis [79].

Another study revealed that the upregulation of miR-342-5p is important for neurogenesis and neuroprotection in an AD mouse model. Downregulation of Ankylin G, a direct target of miR-342-5p, results in AD axonopathy [80]. Liang et al. showed a decrease of miR-153 expression following sAβ treatment of M17 human neuroblastoma cells in combination with H_2_O_2_. APP and APLP2, an APP homologue, are confirmed as direct targets of miR-153, providing additional evidence of microRNA-based regulation of the essential stage of AD progression and the role of oxidative stress in this process [81].

Phosphorylation of Tau protein followed by the accumulation of neurofibrillary tangles is affected by the formation of ROS. Numerous studies confirmed the role of oxidative stress on Tau acetylation and subsequent phosphorylation by GSK-3 kinase or other pathways [82,83,84]. Several microRNAs also contribute to the regulation of Tau phosphorylation. MiR-200a-3p targets BACE1 and PRKACB (catalytic subunit of PKA), reducing Aβ accumulation and Tau hyperphosphorylation, respectively [85]. Li et al. identified overexpressed miR-219 in brains of AD patients. In the SH-SY5Y cell line, miR-219 downregulated Tau phosphorylation by targeting TTBK1 and GSK-3β [86]. GSK-3β alongside with Rbfox1, EP300, and Calpain 2 are directly targeted by miR-132/212, which are among the most downregulated microRNAs in AD [87]. Moreover, Tau mRNA is directly targeted by miR-132/212 [88]. In contrast to the abovementioned cases, overexpression of miR-146b in the AD brain induced abnormal Tau phosphorylation by targeting ROCK1 kinase [89]. Absalon et al. demonstrated a neuroprotective effect of sequence-specific inhibition of miR-26 in primary cortical neurons treated with H_2_O_2_. miR-26 is known to be upregulated in AD patients and contributes to Tau hyperphosphorylation and Aβ accumulation [90].

Screening of AD-associated microRNAs in H_2_O_2_-treated primary hippocampal neurons and a senescent mouse model demonstrated strong upregulation of miR-329, miR-193b, miR-20a, miR-296, and miR-130b. Expression of miR-329 played a critical role in the activity-dependent dendritic outgrowth of hippocampal neurons, whereas miR-130b expressed in the hippocampus was related to chronic stress-induced depression. miR-20a targeted neuronal differentiation markers BCL2, MEF2D and MAP3K12 (ZPK/MUK/DLK), suggesting its key role in the regulation of gene expression during brain development. According to KEGG analysis, upregulated microRNAs participated in the cellular processes closely connected to the occurrence and development of AD, in particular, neurotrophin signaling pathway, MAPK pathway, insulin signaling pathway, and regulation of actin cytoskeleton. This can indicate the importance of abovementioned microRNAs for the development of AD. [91]. miR-330a has also been reported to contribute to alleviation of oxidative stress and mitochondria dysfunction in AD by targeting mRNAs of VAV1, ERK1, JNK1, P38MAPK, and Aβ, which are all upregulated in AD mice, indicating the involvement of the MAPK pathway in AD [92].

The Notch pathway is among important cellular processes that can be associated with oxidative stress [93,94,95]. The Notch-HEY2 pathway in the hippocampal neurons of AD mice was activated with the downregulation of miR-98-5p compared to normal hippocampal neurons. *APP* correlated with levels of miR-98-5p, alongside *HEY2*, *Jagged1*, *Notch1*, *Hes1*, *Hes5*, and *Bax* genes of the Notch pathway, indicating the inhibitory effect of miR-98-5p on these genes and, thus, AD progression. Furthermore, miR-98-5p promoted the growth of hippocampal neurons, inhibited neuronal apoptosis, and improved oxidative stress and mitochondrial dysfunction of AD mice, whereas HEY2 was reported to have opposite effects. These results contradict previous data about promotion of Aβ production by miR-98-5p and upregulation of this microRNA in AD mouse models. Further studies would possibly clarify a more precise role of miR-98 in AD [96].

Despite being affected, microRNAs themselves can trigger oxidative stress in neurons promoting neurodegeneration. miR-125b is known as an important factor of AD progression, promoting APP, BACE1, and Tau overexpression and hyperphosphorylation [97]. In mouse neuroblastoma Neuro2a APPSwe/Δ9 cells, overexpression of miR-125b enhanced oxidative stress by decreasing levels of superoxide dismutase (SOD) together with the stimulation of apoptosis. Additionally, this microRNA stimulates overexpression of TNF-α, IL-1β, and IL-6 inflammatory cytokines, further supporting the connection between inflammation and oxidative stress in degeneration of neurons. Moreover, miR-125b significantly decreased expression of SphK1, which improves memory, learning, and suppresses formation of Aβ peptides [98]. The biological activities of IL2, another inflammatory cytokine, correlate with the JAK/STAT pathway involved in AD development by inducing astrocyte reactivity. A recent study by Wu et al. demonstrated the role of miR-186, a tumor suppressor microRNA, in the downregulation of IL2. Rat brains with decreased expression of miR-186 had been characterized by the elevated levels of IL2, JAK/STAT, Bax, and Cleaved-caspase 3 genes and ROS, whereas BCL2 and SOD activity were downregulated [99].

## 3. Parkinson’s Disease

PD is a common progressive neurodegenerative disorder. PD is primarily characterized by degeneration of dopamine neurons in the substantia nigra pars compacta (SNpc) and their projections to the corpus striatum. Dopamine neuron loss leads to manifestation of PD motor symptoms, such as bradykinesia, resting tremor, postural instability, and rigidity [6]. Additionally, PD patients exhibit a broad range of non-motor symptoms, such as depression, sleep disorders, and dementia. Many of them precede the appearance of motor symptoms and worsen with progression of PD [100].

PD is an age-related disorder, affecting approximately 1% of the population over 60 years old and this number reaches 4-5% in the population over 85-years old. Despite many years of research, mechanisms underlying the pathology of PD are still not well understood. Several genetic mutations associated with PD have been identified and account for at least 5–10% of PD cases; however, in most of the cases, the etiology of PD is unknown [101].

Many different mechanisms have been proposed to drive neuronal death in PD, including oxidative stress. Major sources of oxidative stress in dopamine neurons include dopamine metabolism, mitochondrial dysfunction, impairment of the endogenous antioxidant system, aggregation of the α-synuclein protein, and neuroinflammation (Figure 2) [102].

Selective vulnerability of dopamine neurons suggests a role of dopamine itself in pathogenesis of PD. Normally, dopamine that is newly synthesized or uptaken from the synaptic cleft is removed from the cytosol and stored in synaptic vesicles by vesicular monoamine transporter 2 (VMAT2). Excess of cytosolic dopamine readily oxidizes and forms ROS [6]. MiR-133b indirectly inhibits expression of VMAT2 via downregulation of Pitx3 [103,104]. Therefore, its upregulation may contribute to PD pathology, since dopamine neurons with reduced VMAT2 expression showed increased sensitivity to dopamine-mediated toxicity [105]. Additionally, increased dopamine transporter (DAT)-mediated dopamine uptake may result in oxidative damage and neuronal degeneration [106]. Interestingly, miR-133b can also alter expression of DAT via the same route as VMAT2 [103]. Therefore, decreased levels of miR-133b may result in elevated levels of DAT, contributing to oxidative stress. This suggestion is particularly interesting in the light of findings that miR-133b is downregulated in the midbrain of PD patients [26]. MiR-137 and miR-491 negatively regulate DAT expression and uptake of dopamine by DAT in vitro [107], and decreased expression of these microRNAs may also implicate them in oxidative stress in PD.

Dysfunctional mitochondria is one of the main sources of ROS. Several mutations in genes encoding proteins PINK1, Parkin, and DJ-1 can affect mitochondrial function, increase oxidative stress, and cause autosomal recessive PD in humans [6]. PINK1, together with Parkin, are mitochondrial quality control regulators: they induce disposal of dysfunctional mitochondria reviewed in [108]. PINK1 exhibits a neuroprotective effect in dopamine neurons by inhibiting ROS production [109], while PINK1 knockout in human and mouse dopamine neurons causes increased ROS generation [110]. MiR-27a and miR-27b suppress expression of PINK1 [111], which potentially can induce oxidative stress. Additionally, miR-27a may be implicated in downregulation of mitochondrial complex I subunit NDUFS4 and, together with miR-155, mitochondrial complex V subunit ATP5G3 [112].

DJ-1 is a multifunctional protein and, amongst various roles, it is a regulator of mitochondrial activity and an important player in mediating the oxidative stress response [113,114]. In addition to its role in familial cases of PD, damaged by irreversible oxidation DJ-1 was also reported in the brains of sporadic PD patients [115]. Increased levels of miR-494 downregulate DJ-1 levels and increase cell vulnerability to oxidative stress both in vitro and in vivo [116]. Upregulation of mir-4639-5p, also targeting DJ-1 expression, increases oxidative stress and causes cell death in SH-SY5Y cells, a frequently used dopamine neuron-like model, and its increased expression was reported in PD patients [117]. In addition, miR-34b and miR-34c are downregulated in PD patients (particularly in the SNpc), and their depletion was correlated with mitochondrial dysfunction, increased oxidative stress, and a moderate decrease of SH-SY5Y cell viability. Decreased expression of miR-34b/c was coupled with downregulated expression of Parkin and DJ-1, although mechanism of their action is unclear [118].

The Nrf2-antioxidant response element (ARE) pathway is an endogenous antioxidant system, shown to be downregulated in neurodegenerative diseases. Nrf2 is regulated by Keap1, which facilitates its degradation. Oxidative stress induces translocation of Nrf2 to the nucleus, activating expression of genes, which encode proteins involved in the oxidative stress response, such as SOD1 and GSH (for more details, see [119,120]). miR-7 is capable of repressing Keap1 [121]; what is particularly interesting in the light of this report is that miR-7 is downregulated in the SNpc of PD patients, and its downregulation results in a loss of dopamine neurons in vivo [122]. MiR-153, miR-27a, miR-142-5p, and miR-144 can directly downregulate Nrf2 expression in SH-SY5Y cells [123], potentially contributing to an impaired oxidative stress response.

Histopathologically, PD is characterized by formation of inclusions in neuronal soma (Lewy bodies) or processes (Lewy neurites) with the protein α-synuclein as a major component [124]. Mutations in encoding α-synuclein gene, *SNCA*, and its duplication and triplication were reported to cause familial cases of PD [125]. α-synuclein is capable of inducing oxidative stress and increased levels of ROS, although the exact mechanism is still unclear [126,127,128,129,130]. Multiple microRNAs were reported to control α-synuclein expression, including miR-7, miR-214, miR-153, and miR-34b/c, and their downregulation may contribute to α-synuclein-mediated neurotoxicity in PD [131,132,133,134]. α-synuclein aggregation can also be mediated through its impaired removal by chaperon-mediated autophagy. For example, miR-21, miR-224, miR-373, and miR-379 were demonstrated to downregulate LAMP2 expression, and miR-26b, miR-106a, miR-301b, miR-320a, and miR-16-1 were shown to suppress expression of Hsc70 [135,136,137]. Upregulation of some of these microRNAs were detected in PD patients [120]. MicroRNA regulation of α-synuclein expression has recently been systematically reviewed elsewhere [138]. Altogether, the literature describes multiple mechanisms for microRNAs to contribute to α-synuclein accumulation, which consequently could lead to oxidative stress.

Neuroinflammation, mediated by microglia and to a lesser extent by astrocytes and oligodendrocytes, was shown to play an important role in PD pathophysiology. Particularly, activated microglia can produce numerous cytotoxic substances, including superoxide, and therefore contribute to oxidative stress in the brain (for more details, see [139,140]). Some microRNAs were reported to be implicated in neuroinflammation, such as miR-155 (pro-inflammatory), and miR-146a and miR-124 (anti-inflammatory) [124]. Interestingly, miR-155 was found to be upregulated in an α-synuclein in vivo model of PD and was proposed to mediate α-synuclein-induced inflammation [141,142]. Additionally, increased levels of miR-155 were reported in PD patients. In the same study, downregulation of miR-146a was also demonstrated [143]. MiR-124 attenuates microglia activation and improves survival of dopamine neurons in the MPTP model of PD [144]. PD-associated proteins, including Parkin, DJ-1, and α-synuclein, can induce neuroinflammation by activating microglia [145].

## 4. Amyotrophic Lateral Sclerosis

Amyotrophic lateral sclerosis (ALS) is a neurodegenerative disease characterized by a loss of upper and lower motor neurons in the brain and spinal cord [146], which leads to loss of voluntary control over muscles and subsequent muscle atrophy. Patients gradually experience worsening symptoms of muscle weakness, problems with speaking, chewing and swallowing, and eventually breathing difficulties most often leading to death due to respiratory failure. About one out of 300–500 humans is affected by ALS, with the incidence being higher in men. The risk increases with age and survival is estimated at 3-4 years after onset. ALS presents either in a sporadic or a familial form. There are many genes associated with the familial form and a few mutations which are known to be the cause, the most common ones being on RNA binding protein FUS (*FUS*), TAR DNA-binding protein 43 (*TARDBP*), chromosome 9 open reading frame 72 (*C9orf72*) and Cu^2+^/Zn^2+^ superoxide dismutase (*SOD1*) [147]. Notably, *TARDBP* and *FUS* are involved in RNA biology, including microRNA processing [148]. Non-genetic factors are also implicated in ALS. For instance, environmental insults can cause oxidative stress through the release of free radicals, mainly ROS and reactive nitrogen species, which may lead to epigenetic modifications and changes in gene expression relevant for ALS [149].

Supporting evidence for the role of oxidative stress in ALS was demonstrated by a recent meta-analysis which showed that malondialdehyde, 8-hydroxyguanosine, and Advanced Oxidation Protein Product were significantly elevated in the peripheral blood of ALS patients when compared to controls, as opposed to levels of antioxidant glutathione and uric acid which were downregulated [150]. Other oxidative stress markers such as Cu, SOD, glutathione peroxidase, Co-Q10, and transferrin did not have a link to ALS.

The progressive loss of motor neurons happens relatively fast compared to other neurodegenerative diseases and causes a wide variety of clinical symptoms related to motor deficits, making early diagnosis of ALS challenging. Thus, there is an active search for biomarkers of the disease and microRNAs could represent one option as their expression signatures have been studied in patients. Many studies have identified differential expression of small RNAs, including microRNAs, in the muscle, cerebrospinal fluid, motor neuron progenitors, and blood as well as in *post mortem* tissue samples (spinal cord, brain stem, and the brain) of both sporadic and familial ALS patients compared to healthy controls [151,152,153,154,155]. Besides being valuable as biomarkers, many microRNAs are also studied from a therapeutic point of view as regulating them may provide an option to treat ALS. For example, using an AAV-mediated artificial microRNA targeting *SOD1*, which is involved in reducing ROS and one of the causal genes of ALS, has shown efficient silencing of the gene in macaques [156].

A number of differentially expressed miRNAs in ALS patients versus controls regulate genes involved in oxidative stress, e.g., reducing or counteracting ROS/reactive nitrogen species and may be useful as biomarkers and/or therapeutics. For example, miR-27a, miR-34a, miR-155, miR-142-5p, and miR-338-3p have been studied as biomarkers and potential therapeutic targets in relation to ALS and are involved in oxidative stress directly or indirectly [153,155,157,158] (Figure 3).

miR-34a regulates an X-linked inhibitor of apoptosis (XIAP) that is linked to oxidative stress-induced senescence and Sirtuin 1 (SIRT1), which is protective against oxidative stress-induced apoptosis [154,159]. Of interest, SIRT1 is downregulated in PD [160]. Moreover, ALS patient-derived cell lines have a reduction of miR-34a, which is rescued by treatment with enoxacin, a small-molecule drug stimulating microRNA biogenesis [154]. Thus, enoxacin and other microRNA biogenesis stimulating drugs can potentially be used as ALS therapy [52].

The Nrf2-ARE pathway regulates many genes involved in redox reactions and has been linked to ALS [161]. It is regulated by several microRNAs, directly by e.g., aforementioned miR-27a and miR-34a and indirectly by e.g., miR-7 and miR-494, which regulate Nrf2 modulating proteins [116,121,123,162]. Furthermore, inhibiting miR-142-5p reduces oxidative stress via upregulation of the Nrf2-ARE signaling pathway, and it is downregulated in the CSF of sporadic ALS patients [155,163].

MiR-155 has been shown to be upregulated in both sporadic and familial ALS patients, and inhibiting it in the brains of SOD1G93A mice increases both survival and disease duration [157]. Additionally, miR-338-3p regulates certain subunits of mitochondrial OXPHOS complexes [164] and is also implicated in ALS in human patients and mouse models [158,165]. A broader microRNA dysregulation has also been observed in human ALS patient motor neurons and overexpression of ALS-causing genes *FUS*, *TARDBP*, and *SOD1* seem to inhibit pre-miRNA processing by Dicer. Enhancing Dicer with enoxacin improves neuromuscular function in two separate ALS mouse models [52]. Therefore, a treatment strategy not only taking into account oxidative stress, but also microRNA dysregulation could prove to be useful for ALS patients. However, this and the relationship of microRNAs and oxidative stress should be studied much more carefully before engagement of clinical trials.

## 5. Huntington’s Disease

HD is a relatively rare hereditary disorder with the highest prevalence in the white Caucasian population (about 1:10,000 to 1:20,000) reviewed in [166]. HD is caused by abnormal expansion of a repeated trinucleotide (CAG) sequence in the huntingtin (*HTT*) gene, translated to a long polyglutamine stretch in mutant huntingtin (mHTT) protein or, via repeat associated non-ATG (RAN) translation, to homopolymeric proteins prone to aggregation (for detailed review, see [166,167,168]). Longer CAG repeats correlate with an earlier age of onset of disease symptoms, which include severe motor (chorea, bradykinesia, and dystonia), cognitive (executive function, memory, attention and visuospatial functions) and psychiatric (anxiety, aggression, apathy and depression) disturbances, combined with sleep and circadian disorders, weight loss, skeletal muscle wasting, testicular atrophy, and peripheral immune system alterations [166,168]. The majority of these symptoms are caused by degeneration of striatal GABAergic medium spiny neurons and the cortical neurons projecting to them, accompanied by astrogliosis and microglia activation. Glutamate excitotoxicity, caused by reduced astrocyte glutamate uptake, further exacerbates neurodegeneration. Progressive atrophy of the striatum and cerebral cortex leads to patient death at 15–20 years from the disease onset [166,168].

On a molecular level, HD is characterized by the presence of nuclear inclusions and cytoplasmic aggregates containing mHTT and RAN translation proteins, transcription dysregulation (including large changes in microRNAs), inhibition of proteasome activity and autophagy, defects in synaptic neurotransmission, mitochondrial dysfunction, and oxidative stress [8,166,168,169,170]. A direct link between microRNA dysregulation in HD and oxidative stress has not been evidently described in the literature; however, both are highly relevant for HD as summarized below. Moreover, drawing from research on other neurodegenerative disorders (particularly ALS and PD), it seems plausible that global dysregulation of microRNAs in HD and oxidative stress might form a vicious cycle exacerbating each other and potentially worsening disease progression [51].

Dysregulation of transcription caused by interaction of mHTT with Repressor Element 1 Silencing Transcription Factor (REST) affected, among other targets, the expression of several REST-regulated microRNAs in mouse HD models and, importantly, in *post mortem* cortex samples of HD patients, where upregulation of miR-29a and miR-330 and downregulation of miR-132 was observed [171]. Similarly, analysis of cortical microRNA expression in the brains of patients at different HD stages identified progressive downregulation of miR-9, miR-9*, miR-29b, and miR-124a, whereas, in contrast to the study of Johnson et al. [171], no changes of miR-29a and significant upregulation of miR-132 at late disease stages were observed [172]. Interestingly, both wild-type and mHTT interact with Ago2 and localize to P bodies, suggesting that mHTT can affect Ago2 and, consequently, RISC complex activity in HD [173]. Importantly, recent results confirmed the effect of mHTT on Ago2, demonstrating that aggregation of mHTT, through autophagy impairment, can lead to Ago2 accumulation in a mouse HD model and HD patients and, consequently, to global dysregulation of microRNA levels and activity [174]. mHTT mRNA can also lead to generation of small CAG-repeated RNAs, whose generation and neurotoxic activity depend on Dicer and Ago2, potentially affecting microRNA biogenesis [175]. Thus, both transcription and processing of microRNAs appear to be dysregulated in HD. Indeed, analysis of HD mouse models identified common downregulation of miR-22, miR-29c, miR-128, miR-132, miR-138, miR-218, miR-222, miR-344, and miR-674*, as well as reduced levels of Drosha and Dicer mRNA [176]. In line with these results, microRNA sequencing and differential expression analysis demonstrated deregulation of multiple microRNAs in the frontal cortex and striatum of HD patients [177]. Moreover, because microRNA silencing machinery may be impeded in HD due to Ago2 translocation to stress granules [173,174], observed changes in specific microRNAs should be interpreted with caution as they might not reflect a functional outcome on target mRNA regulation (Figure 4).

The unequivocal cause for HD is the CAG expanse in the *HTT* gene and a higher number of CAG repeats leads to an earlier manifestation of the disease. However, large variations in age of disease onset among individuals with moderate (<55) CAG repeat numbers, together with variations in disease progression, strongly imply genetic and environmental modifiers of the disease which could exacerbate detrimental effects of *mHTT* explaining observed variability [178,179]. Oxidative stress or, conversely, capacity of antioxidant defense systems seem highly plausible as modifiers of HD [8,180]. Markers of oxidative stress rise with transition from the asymptomatic to symptomatic phase in HD patients [181], and oxidative stress is widely described as the main contributor to cell death in HD [8]. While there are no studies specifically addressing the link between microRNAs and oxidative stress in HD, some of the above-mentioned microRNAs, such as miR-9, miR-29, miR-124, and miR-128, changed in HD models and patients, have also been predicted to target genes involved in the oxidative stress response [53]. Similarly, general dysregulation of the microRNA network observed in HD will affect neuronal susceptibility to stress, including oxidative stress [182,183], which putatively could affect pace of disease progression. Conversely, it is tempting to speculate that strategies based on boosting microRNAs processing machinery could slow down demise of neurons in HD similarly to what we and others have shown in models of ALS [52] and PD [30].

Overall, while both oxidative stress and microRNA dysregulation are established features in HD, their interaction remains largely unexplored, yet an intriguing and promising topic for further studies.

## 6. Common and Unique MicroRNAs Affecting Oxidative Stress in Neurodegenerative Diseases

As discussed above, neurodegenerative diseases share many similarities, including mitochondrial dysfunction, formation, and spread of insoluble protein inclusions and, as reviewed here, oxidative stress and deregulation of microRNA networks. Among multiple microRNAs associated with neurodegenerative diseases, we have focused on those implicated in the oxidative stress response (Figure 1, Figure 2, Figure 3 and Figure 4 and Table 1). For many microRNAs, association with oxidative stress was not reported in the original publication, which frequently only demonstrated the change in its level in a selected neurodegenerative condition. In such cases, we consulted other studies, like [53], to identify if a particular microRNA can be involved in the oxidative stress response. Comparison of microRNAs associated with each of the four neurodegenerative diseases reviewed here identified only a small set of common microRNAs affecting the oxidative stress response in different neurodegenerative conditions (Table 1), and no single common oxidative stress-implicated microRNA was reported to be associated with three or four diseases. However, compared to AD and PD, relatively few studies have addressed changes in microRNA levels in ALS and HD, and, therefore, it is reasonable to expect that many more microRNAs associated with these diseases are awaiting their discovery. Nevertheless, many microRNAs are common at least between two neurodegenerative conditions (Table 1); among them are miR-34, miR-124, miR-132, miR-26, mir-7 which are highly expressed in the brain [184,185] and regulate multiple oxidative stress-related pathways (Figure 1, Figure 2, Figure 3 and Figure 4). Such microRNAs are particularly attractive as potential therapeutic targets for the treatment of neurodegeneration. However, the unique microRNAs also deserve attention, as they may be reflecting fundamental differences in the development and progression of particular neurodegenerative disease and serve as specific biomarkers, facilitating and accelerating disease diagnosis.

## 7. Challenges and Perspectives

The above reviewed results clearly demonstrate the intrinsic link between oxidative stress and microRNAs in ageing and disease. However, there are many questions, experimental details, and technical difficulties that need to be solved to bring microRNA-based therapies to clinical use. We undoubtedly have learned a lot about microRNAs and oxidative stress from experiments in cultured cells and extrapolating results from cancer research, but we should exercise caution in translating the findings obtained in cell culture to human neurons. Despite continuous improvement of computational algorithms, prediction and validation of microRNA-mRNA regulation remains challenging [186,187]. Many reported results on microRNA-mRNA regulation are obtained using luciferase reporter assays and transient transfection of microRNA mimics, which are known to cause unspecific general effects on the microRNA biogenesis pathway [188]. The use of proper controls (scrambled microRNAs and reporters with mutated putative binding sites) in such studies is, therefore, absolutely crucial for their validity. Additional caution in interpretation of microRNA overexpression studies should be taken since achieved and functionally effective overexpression levels might be orders of magnitude higher than normally observed.

MicroRNA expression profiles in neurons and glia in vivo are cell type-specific and different from cultured immortalized cells, as are 3′UTR isoforms [189,190], and, moreover, they change with age and the stage of the disease. Furthermore, expression patterns of microRNAs and their putative targets are distinct in different neuronal populations [191]. Thus, ideally, we should address regulation of endogenous mRNA by endogenous microRNAs, for example, by utilizing target protectors introduced to post-mitotic neurons at the lowest possible concentrations, using proper controls [192,193]. Development of new genetic methods, such as CRISPR/Cas9-mediated gene knockout [194,195,196], greatly facilitated loss-of-function genetic studies, enabling relatively easy deletion of both individual microRNAs and whole microRNA families in cultured cells and in vivo [197,198,199,200]. Both knockout and base editing using CRISPR/Cas9 [201,202,203,204] can be further utilized to selectively mutate or create microRNA binding site(s) on 3′UTR of a particular gene, allowing for precisely addressing the consequences of modulation of individual microRNA-mRNA binding. Identified neuroprotective microRNAs can be introduced to the brain using gene therapy vectors, similar to the ones used in clinical trials for neurotrophic factor expression in neurodegenerative disorders [205].

Translation of the results from animal to human settings has long been an issue in neurodegeneration research, with many neuroprotective treatments successfully working in rodent and even primate models, but not in human patients, failing at the stage of double-blinded randomized clinical trials [206]. Neither genetic nor toxin-based rodent models fully recapitulate features of neurodegenerative diseases. While many AD and PD models focus on protein aggregation, other factors contributing to neurodegeneration clearly exist. Mouse and human midbrain progenitors and dopamine neurons have distinct RNA expression profiles and species-specific differences, for example, the presence of neuromelanin and differences in dopamine oxidation [10,207]. Genetic mutations, which lead to early onset familial PD in humans, do not recapitulate the disease when introduced to rodents [208,209,210]. The lack of appropriate neurodegenerative disease models greatly impairs studies of the disease-related microRNAs. While the majority of microRNAs are conserved between rodents and humans, a number of primate- and human-specific microRNAs have been identified [211,212]. Furthermore, existing data demonstrate that some genes may exhibit human-specific regulation by microRNAs [213,214]. These questions have been partly addressed by the analysis of microRNA–mRNA interactions in neurons derived from patients at different stages of disease progression; however, obtaining high quality RNA in sufficient amounts from specific neuronal populations in *post mortem* brain samples is technically very challenging. Development of more sensitive methods, such as single cell microRNA–mRNA co-sequencing [215] would greatly improve the analysis of patient samples. Studies of post mortem tissue samples are also limited in that they only provide a snapshot of microRNAs changed at a particular disease stage, whereas longitudinal studies would have been much more informative.

Fortunately, current advances in differentiation of patient-derived induced pluripotent stem cells towards specific neuronal populations have finally allowed studying neurodegeneration and, particularly, microRNA alterations, in human disease models [216,217]. However, the protocols for human stem cell reprogramming and differentiation are still challenging, and the obtained neurons have embryonic or early postnatal phenotype, rather than adult neurons affected by neurodegeneration in patients. Culturing cells in artificial in vitro environments can affect their mRNA and microRNA expression patterns and oxidative stress levels (for a review of the current state of the field and challenges, see [216]). We are still lacking the methods to reliably detect and monitor levels of oxidative damage in live cells [218]. Development of such experimental techniques and models would also enable longitudinal studies to address the question on whether oxidative stress is a cause or consequence of other processes affecting neuronal survival, such as mitochondrial dysfunction, protein aggregation, or microRNA biogenesis disruption.

Focusing exclusively on neurons will not be sufficient to understand neurodegeneration—astrocytes, oligodendrocytes, and microglia are important players which may also be involved in modulating oxidative stress effects, for example, by regulating neuroinflammation. Therefore, to uncover molecular mechanisms behind human neurodegenerative diseases, we need to study human models representing and recapitulating the interaction between several neural cell types. Three-dimensional human brain organoids offer a great hope for neurodegeneration modeling [219], though it remains to be seen whether such organoids could be developed to the stage mature enough to model properties of the aged or even the adult brain. In this respect, one very promising direction would be to establish humanized animal models based on transplantation of human neural cell precursors to the rodent brain. A similar strategy has already been successfully implemented to obtain humanized mice with brains chimeric for human glia [220]. For example, it has recently been shown that, after transplantation to the rat midbrain, a proportion of human embryonic stem cell-derived neuron precursors will differentiate to nigral dopamine neurons, integrate into appropriate neuronal circuits, and regrow axons to innervate their natural targets [221,222,223]. It is therefore possible in principle to obtain rodents containing human glia, microglia, and neurons correctly differentiated and integrated into host neuronal circuits and use these humanized animals to model degeneration of human neurons in a human-specific cell environment. Clearly, more work is needed to overcome technical and ethical hurdles; however, recent progress in the development of molecular tools and cellular models gives a strong hope that successful treatments to cure neurodegenerative diseases may finally be available.

## Figures and Tables

**Figure 1 ijms-20-06055-f001:**
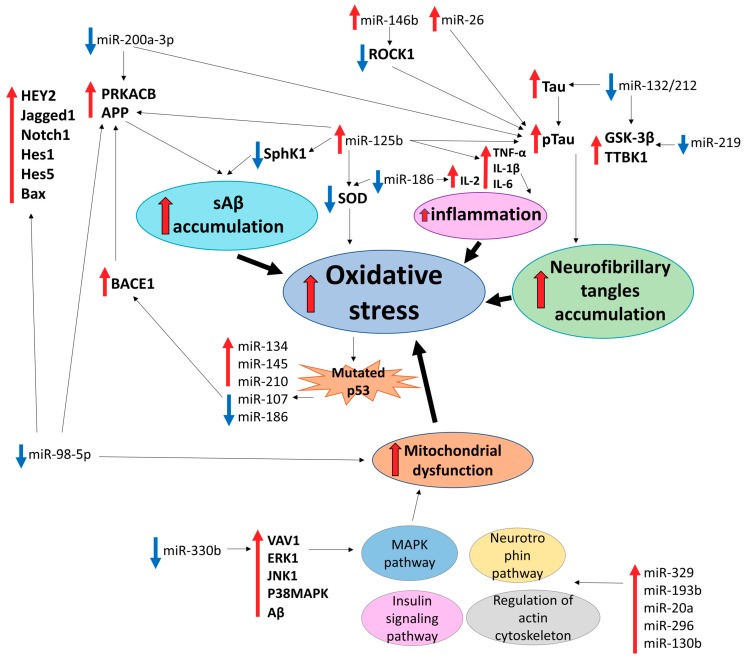
MicroRNAs implicated in oxidative stress-related cellular pathways in Alzheimer’s disease.

**Figure 2 ijms-20-06055-f002:**
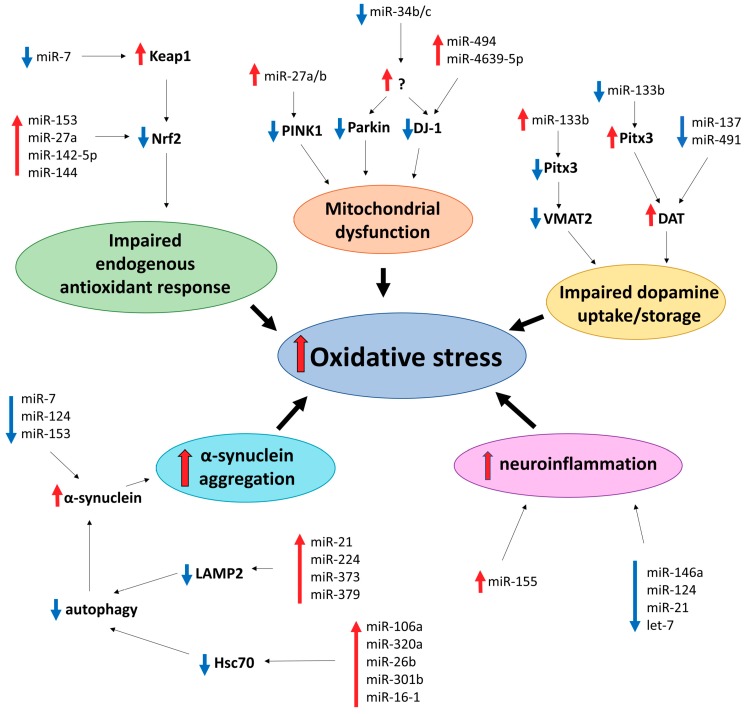
MicroRNAs implicated in oxidative stress-related cellular pathways in Parkinson’s disease.

**Figure 3 ijms-20-06055-f003:**
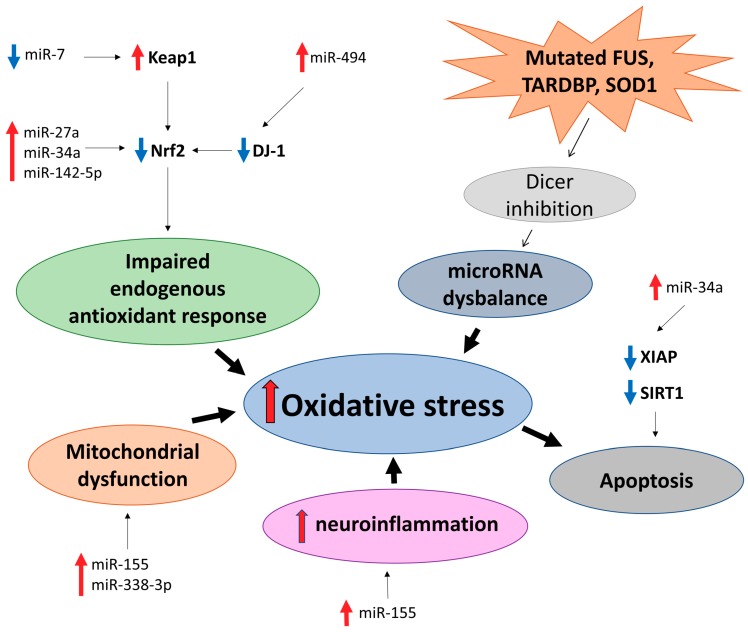
MicroRNAs implicated in oxidative stress-related cellular pathways in Amyotrophic Lateral Sclerosis (AMS).

**Figure 4 ijms-20-06055-f004:**
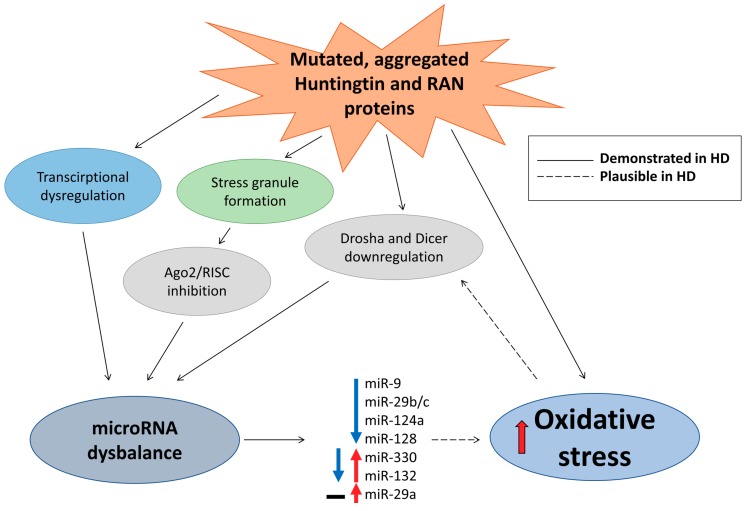
MicroRNAs implicated in oxidative stress-related cellular pathways in Huntington’s Disease.

**Table 1 ijms-20-06055-t001:** MicroRNAs associated with neurodegenerative diseases (AD, PD, ALS, and HD) and implicated in regulation of oxidative stress and related cellular pathways. Bolded are microRNAs associated with more than one neurodegenerative disease.

Disease	Associated microRNAs Involved in Oxidative Stress Regulation	References
Alzheimer’s disease	miR-107	[72,73,78]
	miR-125b	[97]
	miR-130b	[91]
	**miR-132**/212	[87,88]
	miR-134	[72]
	miR-145	[72]
	**miR-146b**	[89]
	**miR-153**	[81]
	miR-186	[79,99]
	miR-193b	[91]
	miR-200a-3p	[85]
	miR-20a	[91]
	miR-210	[72]
	miR-219	[86]
	**miR-26**	[90]
	miR-296	[91]
	miR-329	[91]
	**miR-330a**	[92]
	miR-342-5p	[80]
	miR-98-5p	[96]
Parkinson’s disease	miR-106a	[135]
	**miR-124**	[124,144]
	miR-133b	[103]
	miR-137	[106]
	miR-142-5p	[123]
	miR-144	[123]
	**miR-146a**	[124]
	**miR-153**	[123,133]
	**miR-155**	[124,141]
	miR-16-1	[137]
	miR-214	[132]
	miR-224	[135]
	**miR-26b**	[135]
	**miR-27a/b**	[110,111,123]
	miR-301b	[135]
	miR-320	[136]
	**miR-34b/c**	[118,134]
	miR-373	[135]
	miR-379	[135]
	mir-4639-5p	[117]
	miR-491	[106]
	miR-494	[116]
	miR-7	[121,122,131,133]
ALS	miR-142-5p	[155,163]
	**miR-155**	[157]
	**miR-27a**	[158,165]
	miR-338-3p	[162]
	**miR-34a**	[116,154]
Huntington’s disease	**miR-124a**	[172]
	miR-128	[176]
	**miR-132**	[171,172,176]
	miR-29a/b/c	[171,172,176]
	**miR-330**	[171]
	miR-9	[172]

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
