# Peer review of "Interplay between MicroRNAs and Oxidative Stress in Neurodegenerative Diseases"

_ijms, 2019, doi:10.3390/ijms20236055_

Round 1

Reviewer 1 Report

This review summarizes a role of miRNAs in oxidative stress and related mechanisms in neurodegeneration. The article is well written, comprehensive, and informative. It covers the current literature and the state of the research in this field. Some text passages are quite dense and some parts of the text deviate from the main subject. However, this additional information is integrative in that it helps the reader to further connect the subject to a broader role of miRNAs in neurodegeneration.

A small improvement could be a more sophisticated narrative of AD pathogenesis, as the field is currently moving in different directions. The discussion puts a lot of emphasis on Abeta and ptau pathogenesis which should be viewed more critical, in particular when focusing on oxidative stress and associated metabolic functions. This also pertains to a more critical assessment of the relevance of disease model systems, not only for AD but also for neurodegeneration in general. There are clear lacks in the fields which also affect miRNA studies. I suggest to making some improvements with these points in mind, including a comment in section 7.

There are several mistakes in the text, such as missing words, syntax, etc. which need to be taken care of.

Author Response

We thank the reviewers for their positive evaluation and critical comments on our manuscript. Addressing reviewers’ suggestions, we have revised the text adding and rephrasing the required parts, proofread the manuscript and added three figures demonstrating the connection of different biological pathways affected in AD, ALS and HD due to miRNA dysregulation. The introduced changes are highlighted in the revised text. We hope that you will find the revised manuscript acceptable for publication.

Below, please, find our detailed response.

Reviewer 1

A small improvement could be a more sophisticated narrative of AD pathogenesis, as the field is currently moving in different directions. The discussion puts a lot of emphasis on Abeta and ptau pathogenesis which should be viewed more critical, in particular when focusing on oxidative stress and associated metabolic functions. This also pertains to a more critical assessment of the relevance of disease model systems, not only for AD but also for neurodegeneration in general. There are clear lacks in the fields which also affect miRNA studies. I suggest to making some improvements with these points in mind, including a comment in section 7.

We fully agree with the reviewer’s critical view of the role of Abeta and phospho-Tau in AD pathogenesis. Our initial version was reflecting the relative abundance of the literature focusing on Abeta and phospho-Tau. We have rephrased the corresponding text (lines 106-108) to reflect more critical view and highlight the role of oxidative stress in AD pathogenesis.

We also share reviewer’s critical view of the existing neurodegenerative disease models, which we expressed in section 7; we emphasized the drawbacks of the existing models and the need to develop new ones by including comments in lines 489-491 and 495-496.

There are several mistakes in the text, such as missing words, syntax, etc. which need to be taken care of.

We have the manuscript proofread by native English speaker with scientific writing expertise and introduced the changes throughout the text.

Reviewer 2 Report

Article: “Interplay between microRNAs and oxidative stress in neurodegenerative diseases” by Julia Konovalova et al.

In this manuscript the authors review literature supporting the role of miRNAs in the pathological effect of oxidative stress in neurodegeneration, specifically Alzheimer’s disease, Parkinson’s disease, amyotrophic lateral sclerosis and Huntington’s disease. The review summarizes common and unique miRNAs dysregulated in these neurodegenerative diseases. Except for a few points explained below, I think it is worthy of publication.

Some points to consider:

Paraphrase and do not use direct quotes from references without “”  and cite the correct reference. E.g.

Lines 35-37: “Aging population and increased life expectancy in the developed countries are both leading to the increase in incidence of age-related neurodegeneration” Direct quote from Lai et al 2019, which is not even used as a reference.

Lines 153-155: “the MAPK pathway, the neurotrophin signaling pathway, regulation of actin cytoskeleton, and insulin signaling pathways, all of which are closely related to AD occurrence and development” Direct quote from Zhang et al 2014.

Lines281-283: “chromosome 9 open reading frame 72 (C9orf72 ), Cu2+ /Zn2+ superoxide dismutase (SOD1 ), TAR DNA-binding protein 43 (TARDBP ), and RNA binding protein FUS (FUS ).” Direct quote from Ricci et al 2018.

Lines 291-294: “found that the levels of antioxidant glutathione and uric acid were downregulated in ALS patients. However, other oxidative stress markers including Cu, SOD, glutathione peroxidase, ceruloplasmin, triglycerides, total cholesterol, LDL, HDL, Co-Q10, and transferrin were not linked to ALS.” Direct quote from Wang et al 2019.

Choice of references:

Lane 200: Ref 6 may not be the most appropriate reference.

Lane 102: Ref 61 is a review. Try to cite original work and no other reviews. If you cite reviews mention it in the text. Same for Refs 151, 153, 161 (and many others in the manuscript).

Lanes 300-301: Dysregulated miRNAs in ALS have also been found in postmortem tissue (spinal cord, brain steam and brain).

Lanes 313-315: What is the significance of the statement “ Moreover, ALS patient-derived cell lines have a reduction of miR-34a which is rescued by treatment with enoxacin, a small-molecule drug stimulating microRNA biogenesis” A follow up statement is needed to let the reader know the importance of this finding. 

Common miRNAs. There is emphasis in common miRNAs as potential therapeutic targets for neurodegenerative disease. What about unique miRNAs? These could be biomarkers of each disease, which would accelerate diagnosis.

Challenges and Perspectives:

Dysregulated miRNAs have been analyzed in postmortem tissue. This is also a limitation since at that time neurons and the microenvironment may reflect only a snapshot of which miRNAs are involved in disease onset and progression. iPS-derived cells have their own limitations as well including reprogramming.

Figures:

Figure 1 is very useful to see the connection of the different biological pathways affected in PD due to miRNA dysregulation. Similar schematics for the other 3 disease would be helpful.

Spell out terms used only a few times e.g.:

Line 121: 8-oxoGua

The list of abbreviations is much shorter than the ones shown in the manuscript

The text would benefit if it was written in a more concise form. The use of unnecessary words makes it difficult to read.

Avoid using:

“The authors found…”

“Previous studies have shown that” (you will reference the papers so no need to mention it)

“have been found to be”

“recent results..”

Avoid overusing conjunctive phrases: indeed

 “Respectively”

Author Response

We thank the reviewers for their positive evaluation and critical comments on our manuscript. Addressing reviewers’ suggestions, we have revised the text adding and rephrasing the required parts, proofread the manuscript and added three figures demonstrating the connection of different biological pathways affected in AD, ALS and HD due to miRNA dysregulation. The introduced changes are highlighted in the revised text. We hope that you will find the revised manuscript acceptable for publication.

Below, please, find our detailed response.

Reviewer 2

Paraphrase and do not use direct quotes from references without “” and cite the correct reference. E.g.

Lines 35-37: “Aging population and increased life expectancy in the developed countries are both leading to the increase in incidence of age-related neurodegeneration” Direct quote from Lai et al 2019, which is not even used as a reference.

Lines 153-155: “the MAPK pathway, the neurotrophin signaling pathway, regulation of actin cytoskeleton, and insulin signaling pathways, all of which are closely related to AD occurrence and development” Direct quote from Zhang et al 2014.

Lines281-283: “chromosome 9 open reading frame 72 (C9orf72 ), Cu2+ /Zn2+ superoxide dismutase (SOD1 ), TAR DNA-binding protein 43 (TARDBP ), and RNA binding protein FUS (FUS ).” Direct quote from Ricci et al 2018.

Lines 291-294: “found that the levels of antioxidant glutathione and uric acid were downregulated in ALS patients. However, other oxidative stress markers including Cu, SOD, glutathione peroxidase, ceruloplasmin, triglycerides, total cholesterol, LDL, HDL, Co-Q10, and transferrin were not linked to ALS.” Direct quote from Wang et al 2019.

We thank the reviewer for careful and critical reading of the manuscript. We regret these inadvertent direct quotes from published articles and have rephrased the corresponding sentences.

Choice of references:

Lane 200: Ref 6 may not be the most appropriate reference.

Lane 102: Ref 61 is a review. Try to cite original work and no other reviews. If you cite reviews mention it in the text. Same for Refs 151, 153, 161 (and many others in the manuscript).

We fully agree with the reviewer about the need to cite original articles, and we tried to do this as much as we can when describing the studies of microRNAs regulating oxidative stress and related pathways. We have cited reviews mostly when describing available information of the prevalence and mechanisms of progression of particular neurodegenerative disease. Our choice of references was also influenced by the Assistant Editor recommendation expressed in the invitation email stating that “…References should be up-to-date, i.e., 50% or above are the papers published within recent 5 years.”

In response to reviewer’s comment, we have removed reference 6 in line 215 and explicitly mentioned cited reviews throughout the manuscript.

Lanes 300-301: Dysregulated miRNAs in ALS have also been found in postmortem tissue (spinal cord, brain steam and brain).

We have added this information in lines 321-322.

Lanes 313-315: What is the significance of the statement “ Moreover, ALS patient-derived cell lines have a reduction of miR-34a which is rescued by treatment with enoxacin, a small-molecule drug stimulating microRNA biogenesis” A follow up statement is needed to let the reader know the importance of this finding.

We thank the reviewer for pointing out this unclear part. The importance of this finding is that enoxacin, previously used as FDA approved antibiotic, as well as other drug stimulating microRNA biogenesis may be repurposed as possible ALS treatment. We have added a sentence clarifying this point in lines 341-342.

Common miRNAs. There is emphasis in common miRNAs as potential therapeutic targets for neurodegenerative disease. What about unique miRNAs? These could be biomarkers of each disease, which would accelerate diagnosis.

We thank the reviewer for this essential comment; the emerging role of microRNAs as disease biomarkers is clearly important. We have changed the title of section 6 (line 430) and added corresponding text in lines 449-452 to address this point.

Challenges and Perspectives:

Dysregulated miRNAs have been analyzed in postmortem tissue. This is also a limitation since at that time neurons and the microenvironment may reflect only a snapshot of which miRNAs are involved in disease onset and progression. iPS-derived cells have their own limitations as well including reprogramming.

We have extended our discussion of challenges and perspectives in lines 504-506 and 509-514 to address these critical points.

Figures:

Figure 1 is very useful to see the connection of the different biological pathways affected in PD due to miRNA dysregulation. Similar schematics for the other 3 disease would be helpful.

We fully agree with reviewer’s suggestion and have introduced three additional figures for corresponding diseases.

Spell out terms used only a few times e.g.:

Line 121: 8-oxoGua

The list of abbreviations is much shorter than the ones shown in the manuscript

This, as well as several other abbreviations rarely used in the text, have been spelled out. We have also expanded the list of abbreviations as suggested.

The text would benefit if it was written in a more concise form. The use of unnecessary words makes it difficult to read.

Avoid using:

“The authors found…”

“Previous studies have shown that” (you will reference the papers so no need to mention it)

“have been found to be”

“recent results..”

Avoid overusing conjunctive phrases: indeed

 “Respectively”

We have introduced the required changes throughout the text and had the manuscript proofread by native English speaker with expertise in scientific writing.